# Antifungal Activity of Spent Coffee Ground Extracts

**DOI:** 10.3390/microorganisms11020242

**Published:** 2023-01-18

**Authors:** Daniela Calheiros, Maria Inês Dias, Ricardo C. Calhelha, Lillian Barros, Isabel C. F. R. Ferreira, Chantal Fernandes, Teresa Gonçalves

**Affiliations:** 1CNC—Center for Neuroscience and Cell Biology of Coimbra, 3004-504 Coimbra, Portugal; 2Centro de Investigação de Montanha (CIMO), Instituto Politécnico de Bragança, Campus de Santa Apolónia, 5300-253 Bragança, Portugal; 3Laboratório Associado Para a Sustentabilidade e Tecnologia em Regiões de Montanha (SusTEC), Instituto Politécnico de Bragança, Campus de Santa Apolónia, 5300-253 Bragança, Portugal; 4FMUC—Faculty of Medicine, University of Coimbra, 3004-504 Coimbra, Portugal

**Keywords:** spent coffee grounds (SCG), natural extracts, antifungal activity, *Candida* spp., *Trichophyton* spp.

## Abstract

Coffee is one of the most popular and consumed products in the world, generating tons of solid waste known as spent coffee grounds (SCG), containing several bioactive compounds. Here, the antifungal activity of ethanolic SCG extract from caffeinated and decaffeinated coffee capsules was evaluated against yeasts and filamentous fungi. These extracts had antifungal activity against *Candida krusei*, *Candida parapsilosis*, *Trichophyton mentagrophytes*, and *Trichophyton rubrum*, all skin fungal agents. Moreover, SCG had fungicidal activity against *T. mentagrophytes* and *T. rubrum*. To understand the underlying mechanisms of the antifungal activity, fungal cell membrane and cell wall components were quantified. SCG caused a significant reduction of the ergosterol, chitin, and β-(1,3)-glucan content of *C. parapsilosis*, revealing the synthesis of this membrane component and cell wall components as possible targets of these extracts. These extracts were cytotoxic for the tumoral cell lines tested but not for the non-tumoral PLP2 cell line. The analysis of the phenolic compounds of these extracts revealed the presence of caffeoylquinic acid, feruloylquinic acid, and caffeoylshikimic acid derivatives. Overall, this confirmed the antifungal activity of spent coffee grounds, presenting a potential increase in the sustainability of the life cycle of coffee grounds, as a source for the development of novel antifungal formulations, especially for skin or mucosal fungal infections.

## 1. Introduction

Superficial infections of the skin, nails, hair, and mucosa are the most common fungal diseases in humans, affecting approximately a quarter of the global population [1]. These infections are mainly caused by dermatophytes, specifically, *Trichophyton* spp., *Microsporum* spp., and *Epidermophyton* spp., causing well-known conditions such as athlete’s foot, ringworm of the skin and scalp, and onychomycoses [2,3,4,5]. Mucosal infections of the genital and oral tracts are also extremely common, especially vulvovaginal candidiasis. About 75% of the women have had a vulvovaginal infection caused by *Candida* spp. during their lifetime [1,6]. Although superficial infections do not present a high mortality rate, they are unpleasant and affect the patient’s self-esteem and quality of life [1]. Moreover, heavy skin colonization/skin infections by yeasts are considered risk factors for the development of a life-threatening systemic infection in the case of a decreased immune condition of the host [7]. *Candida* species have been classified as human skin commensals since they are ubiquitous yeasts that can be found on the skin and mucosa normal microbiota. Species such as *Candida albicans*, *Candida tropicalis*, *Candida parapsilosis*, and *Candida orthopsilosis*, found on healthy skin, can become pathogenic [8].

*Trichophyton rubrum* and *Trichophyton mentagrophytes* are the main dermatophyte species responsible for causing onychomycosis, accounting about 90% of the toe infections, followed by non-dermatophyte yeasts, *C. albicans* and non-albicans *Candida* spp. [3].

The widespread use of antifungal drugs used both in human and animal health and in the protection of intensive agriculture are turning antifungal therapies to be less effective, due to the emergence of resistant strains. Moreover, some fungal infections require prolonged treatment leading to selective pressure, with fewer drug-susceptible variants and failure treatments as a consequence [9,10]. This turns essential the discovery of new and effective antifungals. Natural products are rich in bioactive compounds that provide unlimited opportunities for new drug development. It is scientifically proven that plants are enriched with many bioactive secondary metabolites such as saponins, alkaloids, and terpenoids, and other compounds containing physiologically active biochemicals that have great potential to have antifungal properties to be used for the treatment of both human and animal mycoses, but still only a few plants have been scientifically studied for quality, safety, and efficacy assessment [11].

Coffee is one of the most consumed beverages and traded commodities in the world and is also a major contributor to dietary antioxidant intake [12]. *Coffea arabica* and *Coffea canephora var. robusta* are the two main species cultivated for coffee commercial production [12,13,14,15]. Besides being a rich source of antioxidants, coffee also has several bioactive metabolites such as xanthines, phenolics, melanoidins, diterpenes, flavonoids, carotenoids, vitamin precursors, and chlorogenic acids, and its consumption is associated with less risk of developing neurodegenerative diseases and less oxidative stress [12,13,16]. Most of these beneficial properties have been associated with coffee but is not yet clear if these benefits are equally present in caffeinated and decaffeinated coffee. In fact, there is evidence that decaffeinated coffee may have identical properties, pointing to the relevant role of compounds other than caffeine [14,17]. The coffee fruit has been indicated as having antimicrobial activities, including against fungi. In fact, the fruit pulp [18,19,20], the skin [21], and the seed extracts [22,23], either roasted [24] or brewed [25,26], have been demonstrated to have bioactivity against fungi agents of animal/human infection and phytopathogens.

The production of coffee requires many processes. The treatment and processing of coffee cherries, the roasting of coffee beans, and coffee brewing generates huge volumes of biowaste contributing to environmental pollution, approximately six million tons of spent coffee grounds (SCG) per year [16,27]. SCG are residues obtained during the ground coffee brewing and they contain several organic compounds that can be exploited as a source of value-added products since only a few substances are extracted from the brewing process [15]. Despite the brewing, studies have investigated the SCG composition and exploited potential applications. SCG still has a high concentration of polysaccharides, oligosaccharides, lipids, aliphatic acids, amino acids, proteins, alkaloids, phenolics, minerals, lignin, melanoidins, and volatile compounds [16,27]. One of the most abundant phenolic compounds in SCG is chlorogenic acid, described to have many valuable properties such as antioxidant activity, anti-inflammatory, antibacterial, anticarcinogenic, antiviral, and hypoglycemic, which have potential to numerous applications in food, agriculture, and pharmaceutical areas [27,28,29,30]. However, the amount of chlorogenic acids depends on the coffee preparation or decaffeination, bean type, bean roasting, geographical origin, and coffee quality [16,17].

As a food by-product, SCG re-use and valorization reduce large amounts of industrial waste, contributing to a significant impact on sustainability [12,13]. In this context, here, we prepared ethanolic caffeinated and decaffeinated SCG extracts from commercial capsules, determined their phenolic profile, evaluated the antifungal activity against filamentous fungi and yeasts, and evaluated the anti-inflammatory activity on RAW 264.7 mouse macrophage cell lines and the cytotoxicity on tumor and non-tumor cell lines.

## 2. Materials and Methods

### 2.1. Fungal Strains and Growth Conditions

*C. albicans* YP0037, *C. glabrata* YP0937, *C. parapsilosis* YP0515, *Candida krusei* YP0338, *T. rubrum* IMF028, *T. mentagrophytes* IMF029, *Aspergillus niger* IMF005, *Fusarium oxysporum* IMF033, and *Alternaria alternata* IMF030 were obtained from the Clinical Yeast Collection—University of Coimbra (CYCUC) of the Institute of Microbiology of the Faculty of Medicine of the University of Coimbra. The fungal strains of *Aspergillus fumigatus* CBS 500.90 and *Alternaria infectoria* CBS 137.90 CBS were obtained from CBS-KNAW Fungal Biodiversity Centre (Utrecht, The Netherlands).

Yeast strains were grown on Yeast Extract Peptone Dextrose (YPD) agar plates with the following composition: 0.5% yeast extract (Panreac, Barcelona, Spain), 1% peptone (Panreac), 2% agar (VWR, Leuven, Belgium, and 2% glucose (Sigma-Aldrich, St. Louis, MO, USA) (*w*/*v*). Yeasts were inoculated and incubated at 30 °C for 24 h.

Filamentous fungi strains were grown on Potato Dextrose Agar (PDA; Difco). *T. rubrum*, *T. mentagrophytes*, *A. fumigatus*, *A. niger*, and *F. oxysporum* were inoculated and incubated at 30 °C for a week. *A. alternata* and *A. infectoria* were inoculated and incubated at 28 °C for a week, with alternating 16 h light and 8 h dark cycle under a BLB blacklight blue lamp (15 W), as previously described [31].

### 2.2. Coffee Spent Grounds Extracts Preparation

Capsules from caffeinated coffee (Delta Qalidus^®^) and decaffeinated coffee (Delta DeQafeinatus^®^) were purchased in local shops. These capsules were used in appropriate coffee machines to make the regular beverages and the capsules were collected and kept at 4 °C until further use (for no more than 48 h). Controls were made with unused capsules. Extracts were prepared following procedures described by Panusa et al. with modifications [15]. The capsules were opened, and the coffee grounds were dried at 35 °C for 5 days. Six different extracts were prepared by adding 2 g of dried SCG or GC (ground coffee) to 100 mL of the extraction solvent (sterile water or ethanol/water 70:30 and *v*/*v*), performing the following conditions: (1) caffeinated SCG extraction with water; (2) caffeinated SCG extraction with ethanol/water 70:30; (3) caffeinated GC extraction with ethanol/water 70:30; (4) decaffeinated SCG extraction with water; (5) decaffeinated SCG extraction and ethanol/water 70:30; (6) decaffeinated GC extraction with ethanol/water 70:30.

The extracts were placed in a water bath at 60 °C for 30 min under continuous stirring. The extracts were centrifuged at 5000× *g* for 5 min at 4 °C, the supernatant was collected, and the pellet was discarded. The ethanolic extracts were evaporated on a rotatory evaporator until about 20% of the initial volume had evaporated and were filtered by 0.2 µm sterile mixed cellulose ester filters (Whatman, Maidstone, UK).

To determine the extracts concentration, 1 mL of each extract was weighed after drying at 50 °C for 4 days.

### 2.3. Analysis of Phenolic Compounds

Phenolic compounds were analyzed in hydroethanolic, infused and decocted extracts, which were redissolved in methanol/water (80:20 and *v*/*v*) to a final concentration of 10 mg/mL and were filtered using 0.22 μm disposable filter disks. The analysis was performed in a HPLC system (Dionex Ultimate 3000 UPLC, Thermo Scientific, San Jose, CA, USA) coupled with a diode-array detector (DAD, using 280 and 370 nm as preferred wavelengths) and a Linear Ion Trap (LTQ XL) mass spectrometer (MS, Thermo Finnigan, San Jose, CA, USA) equipped with an electrospray ionization (ESI) source. Separation was made in a Waters Spherisorb S3 ODS-2 C18 column (3 µm, 4.6 mm × 150 mm; Waters, Milford, MA, USA). The operating conditions were previously described by Bessada et al., as well as the identification and quantification procedures [32]. The results were given as mg per g of extract.

### 2.4. Screening of the Antifungal Activity In Vitro

#### 2.4.1. Antimicrobial Susceptibility Testing

Antimicrobial susceptibility testing using the microdilution broth technique was performed to determine the minimum inhibitory concentration (MIC) of the extracts, following the standard protocol M27-A2 for yeasts and the M38-A2 protocol for filamentous fungi Clinical and Laboratory Standards Institute (CLSI). 

#### 2.4.2. Minimum Fungicidal Concentration

Minimum fungicidal concentration (MFC) is defined as the lowest drug dilution that can yield a 99–99.5% killing of the fungal agent [33]. In that procedure, 20 µL from suspension of the well in which the inhibitory concentration was determined and from the wells with the two higher following concentrations and from the growth control (extract-free medium) were inoculated in the following culture medium: YPD for yeast and PDA for filamentous fungi, and incubated as described for the microdilution broth assay. The MFC considered in this study was the lowest drug concentration where no growth was observed.

### 2.5. Quantification of Membrane and Cell Wall Components

To study the mechanism of action of the extracts, the membrane and cell wall components of the species that showed growth inhibition were quantified.

#### 2.5.1. Inoculum Preparation

The yeast inoculum was prepared by picking *Candida* spp. colonies of an overnight YPD agar plate. The colonies were suspended in sterile 0.85% saline solution and the OD_600_ was measured.

The suspension was used to inoculate 50 mL of Yeast Malt Extract (YME) with the following composition: 0.4% yeast extract (panreac), 1% glucose (Sigma-Aldrich), 1% malt extract agar (Difco) (*w*/*v*) to a final OD_600_ of 0.05 with and without the extracts B and D at the previously determined MIC. An additional control with itraconazole (0.0157 µg/mL) was performed. The cultures were incubated for 24 h with orbital shaking at 30 °C. The cells were collected by centrifugation at 5000× *g* for 5 min and resuspended in sterile water. The fungal dry weight was determined by drying 1 mL of the cell suspension at 50 °C (triplicates).

The filamentous fungi inocula were prepared by covering four–five days colonies with 1 mL of sterile 0.85% saline and gently probing the colonies with a sterile swab. The suspension resulted in a mixture of spores and hyphal fragments that were left to settle for five minutes. The conidia were counted in a haemocytometer and the concentration was adjusted to a final concentration of 10^5^ spores/mL and 500 µL of the suspension were used to inoculate 100 mL of YME with and without the extracts at the previously determined MIC. A control with itraconazole (0.0157 µg/mL) was also performed. The cultures were incubated for 4 days with orbital shaking at 30 °C. The mycelia were isolated and washed with sterile water and then lyophilized.

#### 2.5.2. Determination of the Ergosterol Contents

The total intracellular sterols were extracted following the alcoholic KOH method described by Breivik and Owades with modifications [34]. The spectral absorption pattern between 220 nm and 300 nm, representative of the ergosterol and 24(28)dehydroergosterol [24(28)-DHE] content of the samples, was obtained. Ergosterol and 24(28)-DHE absorbs at 281.5 nm while only 24(28)- DHE absorbs at 230 nm. Therefore, the ergosterol content was determined by subtracting the amount of 24(28)-DHE (OD_230_) from the total ergosterol plus 24(28)-DHE (OD_281.5_).

Ergosterol content was calculated with the following equations, where F is the dilution factor in ethanol 99.9% and 290 and 518 are the E values (in percent per centimetre) determined for crystalline ergosterol and 24(28)DHE, respectively:$$\%\;\mathrm{ergosterol}+\%\;24\left(28\right)\mathrm{DHE}=\frac{\frac{A281.5}{290}\times F}{\mathrm{dry}\;\mathrm{weight}\;}$$
$$\%24\left(28\right)\mathrm{DHE}=\frac{\frac{A230}{518}\times F}{\mathrm{dry}\;\mathrm{weight}}$$
$$\%\;\mathrm{ergosterol}=\left(\%\mathrm{ergosterol}+\%\;24\left(28\right)\mathrm{DHE}\right)-\%24\left(28\right)\mathrm{DHE}$$

#### 2.5.3. Determination the Cell Wall β-(1,3)-Glucan Content

The content of β-(1,3)-glucan in the cell walls was determined by aniline blue assay as described by Fernandes et al. [35]. The fluorescence was determined with a fluorometer (Spectra Max^®^ Gemini EM, Molecular Devices, San Jose, CA, USA) at 405 nm excitation wavelength and 460 nm emission wavelength. Standard calibration curves were performed using curdlan (Sigma C7821).

#### 2.5.4. Determination the Cell Wall Chitin Contents

The cell wall chitin content was determined by quantifying the glucosamine released by acid hydrolysis of purified cell walls as described before [35]. The OD_520_ was measured using SpectraMax^®^ Plus 384 spectrophotometer. Standard calibration curves were performed using glucosamine solution (Sigma-Aldrich^®^, St. Louis, MO, USA).

### 2.6. Characterization of Morphological and Ultrastructural Changes

The ultrastructural changes induced by the caffeinated and decaffeinated SCG extracts in *C. parapsilosis* and *T. rubrum* were analyzed by Transmission Electron Microscopy (TEM).

To perform TEM analysis, the inoculum was prepared as described previously in section “Quantification of membrane and cell wall components”, except that the mycelia of filamentous fungi was not lyophilized. The samples were fixed with 2.5% glutaraldehyde in 0.1 M sodium cacodylate buffer (pH 7.2). Post-fixation was performed using 1% osmium tetroxide for 1 h. After rinsing with the buffer, the samples were dehydrated in a graded ethanol series (30 to 100%), impregnated, and embedded in epoxy resin (Fluka Analytical^®^). Ultrathin sections (80 nm) were mounted on copper grids (300 mesh) and stained with uranyl acetate 2% (15 min) and 0.2% lead citrate (10 min).

TEM images were obtained using FEI-Tecnai^®^ G2 Spirit Bio Twin^™^ transmission electron microscope at 100 kV.

### 2.7. Cytotoxicity Activity

The following human tumor cell lines were used: AGS (gastric adenocarcinoma), CaCo2 (colorectal adenocarcinoma), MCF-7 (breast adenocarcinoma), and NCI-H460 (lung carcinoma). Non-tumor cell line was also tested: PLP2 (primary pig liver culture). All of them maintained in RPMI-1640 medium supplemented with 10% fetal bovine serum, glutamine (2 mM), penicillin (100 U/mL), and streptomycin (100 mg/mL). The culture flasks were incubated at 37 °C and with 5% CO_2_ under a humid atmosphere. The cells were used only when they had 70 to 80% confluence. A known mass of each of the extracts (8 mg) was dissolved in H_2_O (1 mL), in order to obtain the stock solutions with a concentration of 8 mg/mL. Successive dilutions were made to obtain the concentrations to be tested (0.125–8 mg/mL). Each of the extract concentrations (10 μL) were incubated with the cell suspension (190 μL) of the cell lines tested, in 96-well microplates for 72 h. The microplates were incubated at 37 °C and with 5% CO_2_ in a humid atmosphere, after checking the adherence of the cells. All cell lines were tested at a concentration of 10,000 cells/well. After the incubation period, the cells were treated with TCA (10% *w*/*v*; 100 μL) previously cooled, and plates were incubated for 1 h at 4 °C, washed with water, and after drying, an SRB solution (0.057%, *w*/*v*; 100 μL) was added, left to stand at room temperature for 30 min. To remove non-adhered SRB, plates were washed three times with a solution of acetic acid (1% *v*/*v*) and placed to dry. Finally, an adhered SRB was solubilized with Tris (10 mM, 200 μL) and the absorbance at a wavelength of 540 nm was read in the Biotek ELX800 microplate reader. A non-tumor cell culture (PLP2) was also used. This cell culture was prepared from a freshly harvested porcine liver obtained from a local slaughterhouse and submitted to the same procedure as the remaining cell cultures. The results are expressed in terms of the concentration of extracts with the ability to inhibit cell growth by 50%—GI_50_. As a positive control it was used an ellipticine.

### 2.8. Anti-Inflammatory Activity

The extracts were dissolved in H_2_O in order to obtain a final concentration of 8 mg/mL. Successive dilutions were carried out, obtaining the concentrations to be tested (0.125–8 mg/mL). The RAW 264.7 mouse macrophage cell line, obtained from DMSMZ-Leibniz-Institut DSMZ-Deutsche Sammlung von Mikroorganismen und Zellkulturen GmbH, was grown in DMEM medium, supplemented with heat-inactivated (SFB) fetal serum (10%), glutamine and antibiotics, and kept in an incubator at 37 °C, with 5% CO_2_ under a humid atmosphere. Cells were detached with a cell scraper. An aliquot of the cell suspension of macrophages (300 μL) with a cell density of 5 × 10^5^ cells/mL and with a proportion of dead cells below 5% according to the trypan blue exclusion test, was placed in each well. The microplate was incubated for 24 h under the conditions previously indicated, in order to allow an adequate adherence and multiplication of the cells. After that period, the cells were treated with different concentrations of extracts (15 μL and 0.125–8 mg/mL) and incubated for one hour, with the range of concentrations tested being 6.25–400 μg/mL. Stimulation was performed with the addition of 30 μL of the lipopolysaccharide solution—LPS (1 mL/mL), and incubated for an additional 24 h. Dexamethasone (50 mM) was used as a positive control and samples in the absence of LPS were used as a negative control. Quantification of nitric oxide was performed using a Griess reagent system kit (nitrophenamide, ethylenediamine, and nitrite solutions) through the nitrite calibration curve (100 mM sodium nitrite at 1.6 mM) prepared in a 96-well plate. The nitric oxide produced was determined by reading absorbances at 540 nm (ELX800 Biotek microplate reader, Bio-Tek Instruments, Inc., Winooski, VT, USA) and by comparison with the standard calibration line (y = 0.0066x + 0.1349; R^2^ = 0.9986). The results were calculated through the graphical representation of the percentage of inhibition of nitric oxide production versus the sample concentration and were expressed in relation to the concentration of each of the extracts that causes the 50% inhibition of nitric oxide production—IC_50_.

### 2.9. Statistical Analysis

Data analyses were performed using a GraphPad Prism software (Version 8.0.1, Dr. Harvey Motulsky, San Diego, CA, USA). Data are presented as means ± SEM. Statistical differences were determined using one-way or two-way ANOVA, with Dunnett’s post-hoc test for multiple comparison test and T-student test. Results with *p* < 0.05 were considered statistically significant.

## 3. Results

### 3.1. Extract Preparation and Total Phenol Analysis

Six extracts were prepared by adding 2 g of dried SCG or GC, caffeinated or decaffeinated, to 100 mL of the extraction solvent: water or ethanol/water 70:30 (*v*/*v*). Six conditions were tested. The yield of the extraction was 1.4% for caffeinated SCG extraction with water; 1.6% for caffeinated SCG extraction with ethanol/water 70:30; 35.4% for caffeinated GC extraction with ethanol/water 70:30; 1.9% for decaffeinated SCG extraction with water; 2.1% for decaffeinated SCG extraction and ethanol/water 70:30; and 28.3% decaffeinated GC extraction with ethanol/water 70:30.

The phenolic compounds profile of the ethanol caffeinated and decaffeinated SCG extracts were analyzed by HPLC-MS. The results were given as mg per g of extract and are presented in Table 1. The caffeinated SCG extract is rich in trans 5-*O*-caffeoylquinic acid, 4-*O*-feruloylquinic acid, caffeoylshikimic acid, and *cis* 5-*O*-caffeoylquinic acid. The decaffeinated SCG extract has more *cis* 5-*O*-caffeoylquinic acid and caffeoylshikimic acid than caffeinated SCG.

### 3.2. Antifungal Activity

Antimicrobial susceptibility testing was performed to determine the MIC of the SCG and GC extracts, caffeinated and decaffeinated, following standardized CLSI protocol M27-A2 for yeasts and M38-A2 for filamentous fungi. The aqueous extracts of SCG did not show growth inhibition in all the species tested, so it was not further used during the present work. Although the aim of this work was to study the antifungal activity of SCG, we used as a control the same coffee grounds, same origin, and same coffee capsules, but not previously brewed. The objective of using GC was to verify if the commercial coffee grounds inside the capsules had an antifungal effect, and if the brewing, with the recommended coffee machine, eliminated the bioactivity. The results showed that GC had a stronger fungal growth inhibition than SCG, with lower MIC values when compared with SCG extracts (Table 2). The species *C. krusei, C. parapsilosis*, *T. mentagrophytes*, and *T*. *rubrum* are the most susceptible to the tested extracts. The SCG did not show antifungal activity in the following species: *A. alternata*, *A. infectoria, A. fumigatus*, *A. niger*, *C. albicans, C. glabrata*, and *F. oxysporum*, so the following studies only proceeded on the most susceptible species. The most susceptible species to decaffeinated extracts were also *C. krusei, C. parapsilosis*, *T. mentagrophytes*, and *T*. *rubrum.* While the decaffeinated SCG MIC values were higher than the MIC values for caffeinated SCG, the opposite was obtained for the GC extracts (Table 2).

Although the CLSI M38-A2 guidelines do not define breakpoints for mold testing, working breakpoints were assigned for analytical purposes. Accordingly, the MIC determined for all the species tested are within the range of MIC breakpoints defined to be susceptible species, except for *A. fumigatus*, which is considered intermediate, and *A. niger* and *F. oxysporum*, which are considered resistant. To understand whether the effect of the extracts is fungicidal or fungistatic against the most susceptible species, the MFC was determined for the SCG extracts. Both SCG extracts were not fungicidal in *C. krusei* and *C. parapsilosis* but were fungicidal in *T. mentagrophytes* and *T. rubrum* (MFC = 137.50 μg/mL of caffeinated SCG and MFC = 300 μg/mL and 150 μg/mL of decaffeinated SCG to *T. mentagrophytes* and *T. rubrum*, respectively (Table 3).

### 3.3. Cell Membrane and Cell Wall Components Modulation in Response to SCG

To elucidate a possible mechanism of action underlying the antifungal effect, ergosterol, β-(1,3)-glucan, and chitin contents of *C. krusei*, *C. parapsilosis*, *T. mentagrophytes*, and *T. rubrum* were determined after exposure to MIC of the SCG extracts (caffeinated and decaffeinated) and were expressed in a percentage in comparison to the non-treated fungal cells (control 100%). Ergosterol was also quantified after exposure to 0.016 μg/mL of itraconazole as a positive control. A lower concentration than the MIC value was used to allow fungal growth and to have enough biomass to extract and quantify ergosterol.

As seen in Figure 1, no significant differences in the ergosterol, chitin, or β-(1,3)-glucan after exposure of *C. krusei* to caffeinated or decaffeinated SCG extracts were detected. Yet, in *C. parapsilosis*, the caffeinated SCG extract reduced the ergosterol content by 49.77% with statistical significance (*p* < 0.01), similar to the reduction (52.32%) caused by itraconazole (*p* < 0.001). This extract also led to a statistically significant reduction of 14.83% of the amount of β-(1,3)-glucan (*p* < 0.01), and a reduction of 21.66% in the chitin content (*p* < 0.05). In *C. parapsilosis*, exposure to the decaffeinated SCG extract only caused a decrease in the chitin content by 38.62%, with statistical significance (*p* < 0.01).

In respect to filamentous fungi, the ergosterol content of *T. mentagrophytes* after exposure to caffeinated and decaffeinated SCG extracts showed no decrease, even when treated with itraconazole (0.016 μg/mL) (Figure 2A). Treatment with caffeinated SCG extracts reduced the total amount of ergosterol of *T. rubrum* by 12.05% but the decaffeinated SCG extracts had no impact. Itraconazole lowered the ergosterol content by 26.68%, with no statistical significance. The β-(1,3)-glucan content of *T. mentagrophytes* and *T. rubrum* (Figure 2B) had no decrease when exposed to caffeinated SCG extracts. The decaffeinated SCG extract only reduced *T. mentagrophytes* β-(1,3)-glucan by 18.75% but not with statistical significance. The quantification of chitin content of *T. mentagrophytes* revealed no significant alterations after treatment with caffeinated and decaffeinated SCG extracts (Figure 2C). The chitin levels of *T. rubrum* decreased by 12.16% after exposure to the caffeinated SCG extract, without statistical significance. Curiously, exposure to the decaffeinated SCG extract increased the total amount of chitin by 25.69% (*p* < 0.05).

### 3.4. Fungal Ultrastructural Modifications in Response to SCG Evaluated by TEM

Considering the modulation caused by the caffeinated SCG extract in the cell membrane and cell wall of *C. parapsilosis* and the potential fungicidal effect on *T. rubrum*, both these species were selected to elucidate the effect of caffeinated and decaffeinated SCG extracts at the ultrastructural level in comparison to untreated fungal cells. Fungal cells were exposed to a MIC concentration of both the extracts and a TEM study was performed. 

*C. parapsilosis* untreated cells (Figure 3A,B) exhibited a well-defined surface and a regular and integral cell wall. However, there appear to be some structures that cross the cytoplasmic membrane and the cell wall to the outside (black arrows). *C. parapsilosis* treated with caffeinated SCG extracts (Figure 3C–E) and decaffeinated SCG extracts (Figure 3F–H) have a thinner (Figure 4) cell wall and more electrodense cell wall (a more electrodense line in the middle of the cell wall), and the same unknown structures that cross the cytoplasmic membrane are present but higher in number (black arrows).

Regarding *T. rubrum*, untreated cells (Figure 5A,B) have a smooth surface, a regular cell wall, a structured and regular organelles disposition, and visible mitochondria with well-defined cristae. In contrast, cells treated with the caffeinated SCG extract (Figure 5C–E) and decaffeinated SCG extract (Figure 5F–H) have organelle disorganization, especially in what concerns mitochondria, an increased number of and larger vacuoles, and there appear to be endomembrane systems suggesting autophagic structures.

To determine whether the SCG extract was associated with antiproliferative activities, its cytotoxicity was assessed using assays in human tumor cell lines AGS (gastric adenocarcinoma), CaCo2 (colorectal adenocarcinoma), MCF-7 (breast adenocarcinoma), and NCI-H460 (lung carcinoma), along with non-tumor cell lines PLP2 (primary pig liver culture). As depicted in Table 4, the SCG extracts exhibited cytotoxic activity in all cancer cell lines tested, at relatively low concentrations. Among the tumor cell lines, the gastric adenocarcinoma human (AGS) was more susceptible to the antiproliferative effects of SCG extracts with GI_50_ values of 55 and 52 µg/mL, for caffeinated and decaffeinated SCG extracts, respectively. Whereas in the normal cell line PLP2, SCG extracts exhibited little cytotoxicity with GI_50_ value > 400 µg/mL, compared to the control, ellipticine. These results are indicative of the potential of SCG extracts as an inhibitor of the growth of human cancer cell lines, especially AGS.

The anti-inflammatory activity of SCG extracts was evaluated by measuring the percent of inhibition of NO production, expressed in IC_50_ in LPS-activated RAW 264.7 cells. Cells were pre-treated with SCG extracts and incubated for 1 h, stimulated with 1 µg/mL of LPS, and incubated for 24 h. As shown in Table 4, the SCG extracts showed inhibitory effects on the NO production in LPS-activated RAW 264.7 with a IC_50_ > 400 µg/mL, which means that the NO production was not inhibited by the SCG extracts, so no anti-inflammatory activity of the SCG was obtained.

## 4. Discussion

The incidences of fungal infections have been increasing over the years and antifungal therapy is still limited. To overcome antifungal resistance, there is an urgent search for alternatives to conventional antifungal drugs; new and novel strategies with high antifungal potential are needed. SCG have been reported as a potential source of bioactive compounds with beneficial health effects. Caffeine is the most abundant compound, followed by chlorogenic acids and derivatives such as caffeoylquinic acids, feruloylquinic acids, *p*-coumaroylquinic acids, and mixed diesters of caffeic and ferulic acids with quinic acid, all described as powerful in vitro antioxidants [16,27,28,36].

In the present study, SCG from caffeinated and decaffeinated coffee were obtained from Delta Qalidus^®^ and Delta deQafeinatus^®^ capsules (Portuguese brand of coffee products), respectively. According to Delta Q^®^ commercial online information, Delta Qalidus^®^ is a fusion of arabica coffee beans from Honduras and robusta coffee from Angola and Cameroon while Delta deQafeinatus^®^ is a mix of arabica coffee from Brazil and robusta coffee from Vietnam and Uganda. Aqueous and ethanolic extracts were prepared and microdilution broth was performed to study the susceptibility of common fungal agents of human infections. Only the ethanolic extracts showed antifungal activity. Mussato et al. quantified the total phenolic compounds and antioxidant activity of SCG extracts with different extraction solvents and obtained a higher yield using methanol in comparison with water, justifying that phenolic compounds are usually more soluble in organic solvents that are less polar than water [27]. Extraction with ethanol as the solvent recovered more than 90% of the phenolic compounds present in SCG [15,36].

The ethanolic caffeinated SCG extract obtained in the present study is abundant in phenolic compounds, namely in the chlorogenic acids derivatives, as trans 5-*O*-caffeoylquinic acid, 4-*O*-feruloylquinic acid, caffeoylshikimic acid, and *cis* 5-*O*- caffeoylquinic acid. The decaffeinated SCG extract is also abundant in the chlorogenic acid derivatives *cis* 5-*O*-caffeoylquinic acid and caffeoylshikimic acid. Chlorogenic acid and its derivatives have already been reported to have antifungal activity against *C. albicans* by impairing ergosterol biosynthesis and disrupting the cell membrane [37]. Another study showed the antifungal potential of chlorogenic acids against phytopathogenic fungi causing early membrane permeabilization of the spores [38]. Ma and Ma showed the antifungal activity of chlorogenic acid derivatives by inhibiting glucan synthase leading to a decrease in glucan content and cell wall disruption, suggesting their potential as antifungal agents [39]. The antifungal activity of 5-*O*-caffeoylquinic acid against *Candida* sp. and *Aspergillus* sp. has also been reported before [40,41].

SCG from caffeinated and decaffeinated coffee extracts showed antifungal activity against the same fungal species: *C. krusei*, *C. parapsilosis*, *T. mentagrophytes*, and *T. rubrum*, fungi associated with fungal skin infections and not to *A. alternata, A. infectoria, A. fumigatus, A. niger, C. albicans*, or *C. glabrata* fungi more associated with other human pathologies. This could suggest an interest for these extracts for topical application. Interestingly, the MIC determination of SCG from caffeinated and decaffeinated coffee extracts revealed to be fungicidal to *T. mentagrophytes* and *T. rubrum*, killing 99.9% of the fungal agents. The control using ground coffee (GC) extracts revealed that unused coffee grounds had antifungal activity, with lower MIC values. The brewing process of GC removes water-soluble compounds, making SCG less concentrated. Nonetheless, SCG extracts have antifungal activity against the same species as GC extracts. Despite being less concentrated, the bioactive compounds with antifungal activity remain present in SCG. Previous studies have highlighted the antimicrobial activity of SCG, mostly with a focus on bacteria [42,43,44,45,46,47]. Studies describing the antifungal effect of SCG are scarce. Several yeast species (*C. krusei*, *C. parapsilosis*, and *C. albicans*) have already been reported to be susceptible to SCG extracts [12,43]. The antifungal activity against filamentous fungi was also recently reported for a SCG extract obtained using isopropanol [45]. However, it is difficult to compare between studies due to different methodologies (type of coffee, extraction method, antifungal susceptibility tests, and fungal strains). Regardless, this is the first study describing the antifungal effect of SCG against fungal agents of dermatophytosis, *T. mentagrophytes*, and *T. rubrum*.

To identify the mechanism of action underlying the antifungal effect of the extracts, cell membrane and cell wall components were quantified. Ergosterol is the main sterol present in the fungal cell membrane, and it is responsible for the membrane’s structural integrity, fluidity, and permeability [48,49,50]. Chitin and glucan are fundamental in maintaining fungal cell integrity during growth and adaptation to stress [48,51,52]. Impairment in these components could lead the cell membrane or cell wall to disruption and consequently, cell death [45,50,51,52]. Most antifungal therapy available targets these components; however, conventional antifungals are associated with many side effects and drug to drug interactions, with hepatotoxicity, nephrotoxicity, and myelotoxicity being the most severe side effects [53,54,55].

Quantification of ergosterol, β-(1,3)-glucan, and chitin after exposure to the MIC of SCG from caffeinated and decaffeinated coffee extracts was performed. SCG caffeinated extracts induced significant impairment on ergosterol, β-(1,3)-glucan, and chitin contents of *C. parapsilosis*, while SCG from decaffeinated coffee, caused a non-significant reduction in ergosterol and β-(1,3)-glucan levels but a significant decrease in chitin content. Those might be the targets of these extracts in *C. parapsilosis.* However, it remains unknown where the ergosterol and chitin pathway might be affected by exposure to the extracts. Yet, in *C. krusei*, none of these contents were affected significantly by both extracts, although this species is more sensitive to both these extracts, indicating that another vital structure or physiological pathway might be impaired.

The decrease in the β-(1,3)-glucan and chitin content of *C. parapsilosis* and the thinner and more electrodense cell wall observed on TEM images may suggest a cell wall remodeling and changes to the chitin and β-(1,3)-glucan architecture after exposure to SCG extracts [56]. No scientific explanation was found to justify the structures that cross the cell wall present in *C. parapsilosis* un-treated cells, although treatment with SCG extracts seems to increase the number of these structures, which might be a response to stress induced by the extracts.

Although SCG from caffeinated and decaffeinated extracts were determined to be fungicidal against *T. mentagrophytes* and *T. rubrum*, the quantification of the cell membrane and cell wall components had no significant impact, except on the chitin content of *T. rubrum* when exposed to SCG from decaffeinated extracts. Curiously, chitin biosynthesis increased. Chitin build-up is generally associated with a compensatory mechanism triggered in response to cell wall perturbing agents as a defense mechanism [31,52,57]. According to these results, it was not possible to conclude a target underlying the antifungal activity of both extracts against these dermatophyte species. However, TEM images of *T. rubrum* treated with SCG extracts show disorganized, undefined, and perhaps dysfunctional mitochondria, in contrast to non-treated cells where it is possible to observe healthy, organized, and well-defined mitochondria. Mitochondria are related to antifungal tolerance; its dysfunction is associated with the susceptibility and resistance to cell membrane targeting drugs [58]. Besides that, after exposure to the extracts, it also possible to observe in *T. rubrum* TEM images an increase in the number and size of vacuoles and the presence of structures that resemble endomembrane systems, suggesting that autophagy is activated as a protective mechanism, possibly the mechanism behind the antifungal activity of SCG extracts.

In this study, we also reported that the SCG caffeinated and decaffeinated coffee extracts were cytotoxic for different cancer cell lines, without causing cytotoxicity on the non-tumoral cell line used. This is in accordance with previous works that reports a moderate cytotoxicity of SCG extracts on liver cancer cells (HepG2), which are also lower than the positive control used in the assay [45]. However, our results did not show anti-inflammatory activity of SCG from caffeinated and decaffeinated coffee extracts on the RAW 264.7 cell line. A recent publication reported that the SCG water extracts are more effective than extracts obtained with solvents [59]. Nevertheless, when considering the antifungal activity of the extracts obtained in this study in fungi agents of human skin infections together with the absence of cytotoxicity on a non-tumoral cell line, it can be anticipated as a possible utilization in the therapy of dermatomycosis.

## 5. Conclusions

Coffee consumption generates tons of solid waste SCG, containing several bioactive compounds with potential uses in the pharmaceutical, food, and cosmetic industries [16]. In this study, we show that the ethanolic SCG extracts from both caffeinated and decaffeinated coffee are cytotoxic for different cancer cell lines, indicating antiproliferative bioactivity, with no cytotoxicity on a non-tumoral cell line. Our results also clearly demonstrate that these extracts have a strong antifungal bioactivity, especially against fungi involved in skin infections. This indicates that a safe formulation of an SCG-based phytotherapeutic for topic administration could be an economic and eco-friendly alternative to treat skin fungal infections, promoting a coffee by-product and minimizing waste.

## Figures and Tables

**Figure 1 microorganisms-11-00242-f001:**
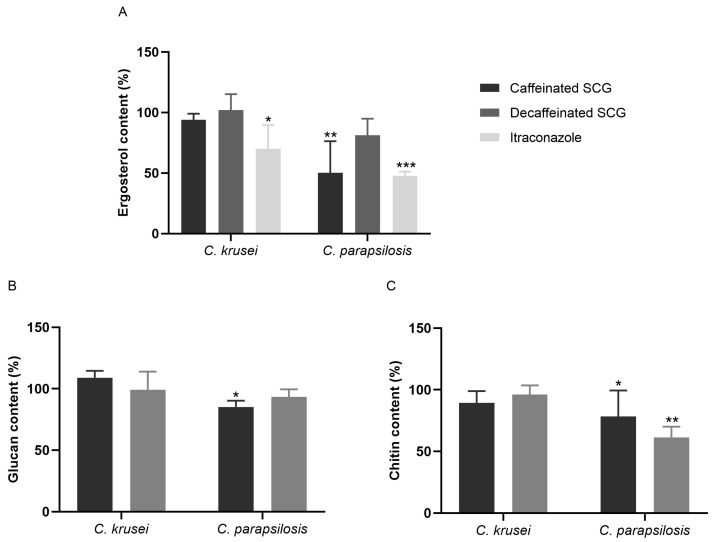
Effects of caffeinated and decaffeinated SCG extracts on cell membrane and cell wall components of yeasts. *C. krusei* treated with 137.50 μg/mL of caffeinated SCG extract, 150 μg/mL of decaffeinated SCG extract, and 0.016 μg/mL of itraconazole. *C. parapsilosis* treated with 275 μg/mL of caffeinated SCG extract, 150 μg/mL of decaffeinated SCG extract, and 0.016 μg/mL of itraconazole. (**A**) Ergosterol, (**B**) β-(1,3)-glucan, and (**C**) chitin contents. Results are expressed as mean ± standard error and are a representation of three independent experiments performed in triplicate, * *p* < 0.05, ** *p* < 0.01, and *** *p* < 0.001 (Dunnett’s post-hoc test). Data were normalized to control values.

**Figure 2 microorganisms-11-00242-f002:**
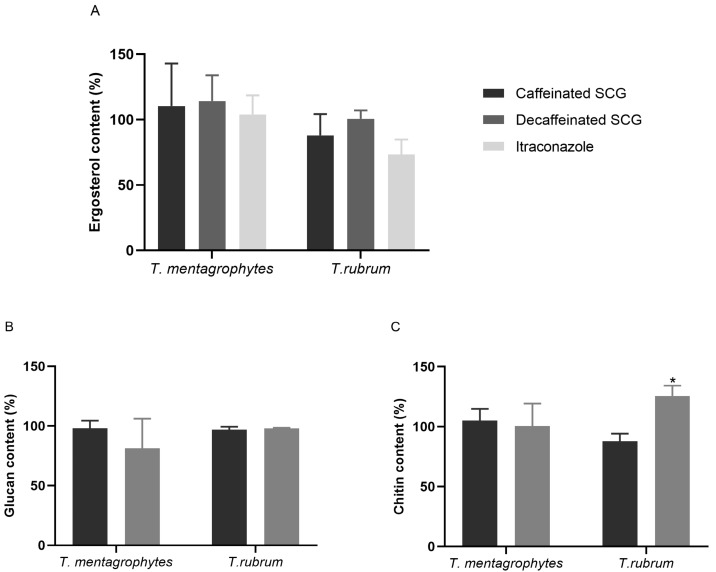
Effects of caffeinated and decaffeinated SCG extracts on cell membrane and cell wall components of filamentous fungi. *T. mentagrophytes* treated with 137.5 μg/mL of caffeinated SCG extract, 300 μg/mL of decaffeinated SCG extract, and 0.016 μg/mL of itraconazole. *T. rubrum* treated with 137.5 μg/mL of caffeinated extract, 150 μg/mL of decaffeinated extract, and 0.016 μg/mL of itraconazole. (**A**) Ergosterol, (**B**) β-(1,3)-glucan, and (**C**) chitin contents. Results are expressed as mean ± standard error and a representation of three independent experiments performed in triplicate, * *p* < 0.05 (Dunnett’s post-hoc test). Data were normalized to control values.

**Figure 3 microorganisms-11-00242-f003:**
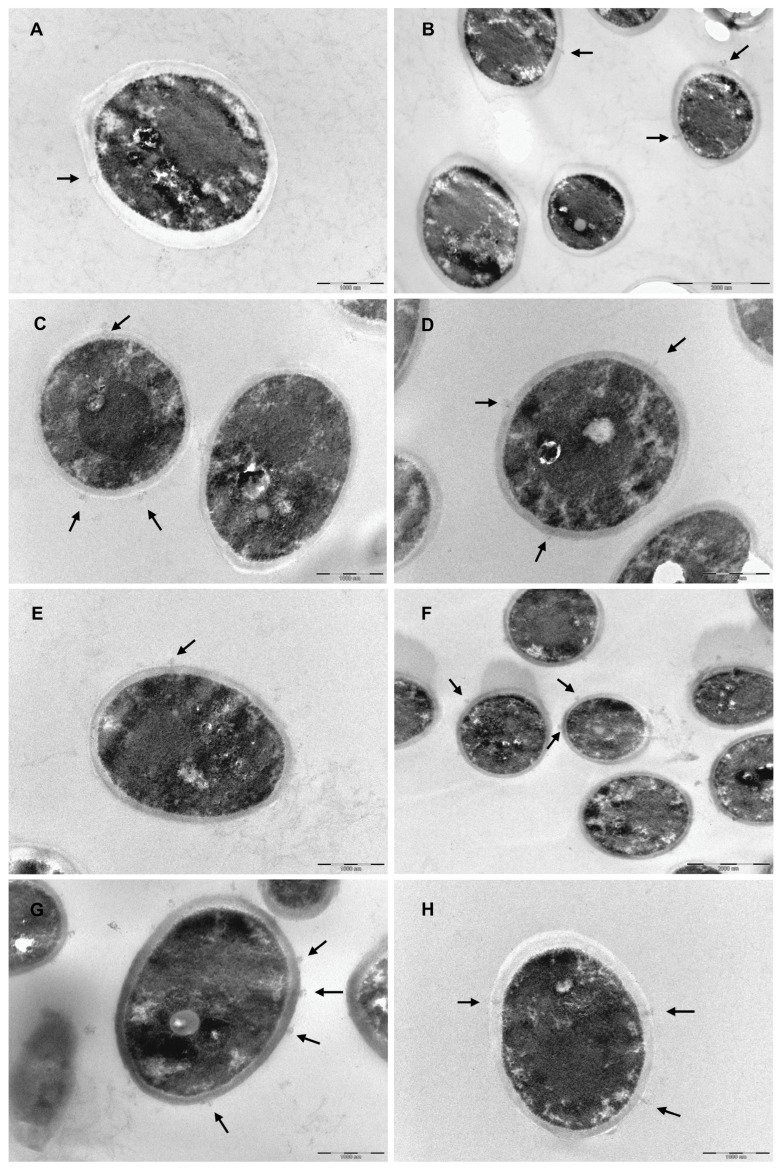
Transmission electron microscopic observations of *C. parapsilosis*. (**A**,**B**) non-treated fungal cells, (**A**) scale bar 1000 nm, (**B**) scale bar 2000 nm; (**C**–**E**) treated with caffeinated SCG extract (275 μg/mL), scale bar: 1000 nm; (**F**–**H**) treated with decaffeinated SCG extract (150 μg/mL), (**F**) scale bar: 1000 nm, and (**G**,**H**) scale bar: 2000 nm. Black arrows indicate structures crossing the membrane and cell wall.

**Figure 4 microorganisms-11-00242-f004:**
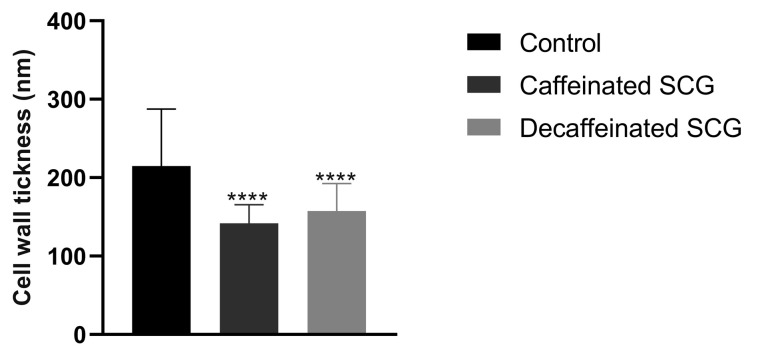
Measurement of *C. parapsilosis* cell wall thickness of non-treated fungal cells (control), treated with caffeinated SCG extract (275 μg/mL), and treated with decaffeinated SCG extract (150 μg/mL). Results are the mean ± the standard error of the mean of one experiment, **** *p* < 0.0001 (Dunnett’s post-hoc test).

**Figure 5 microorganisms-11-00242-f005:**
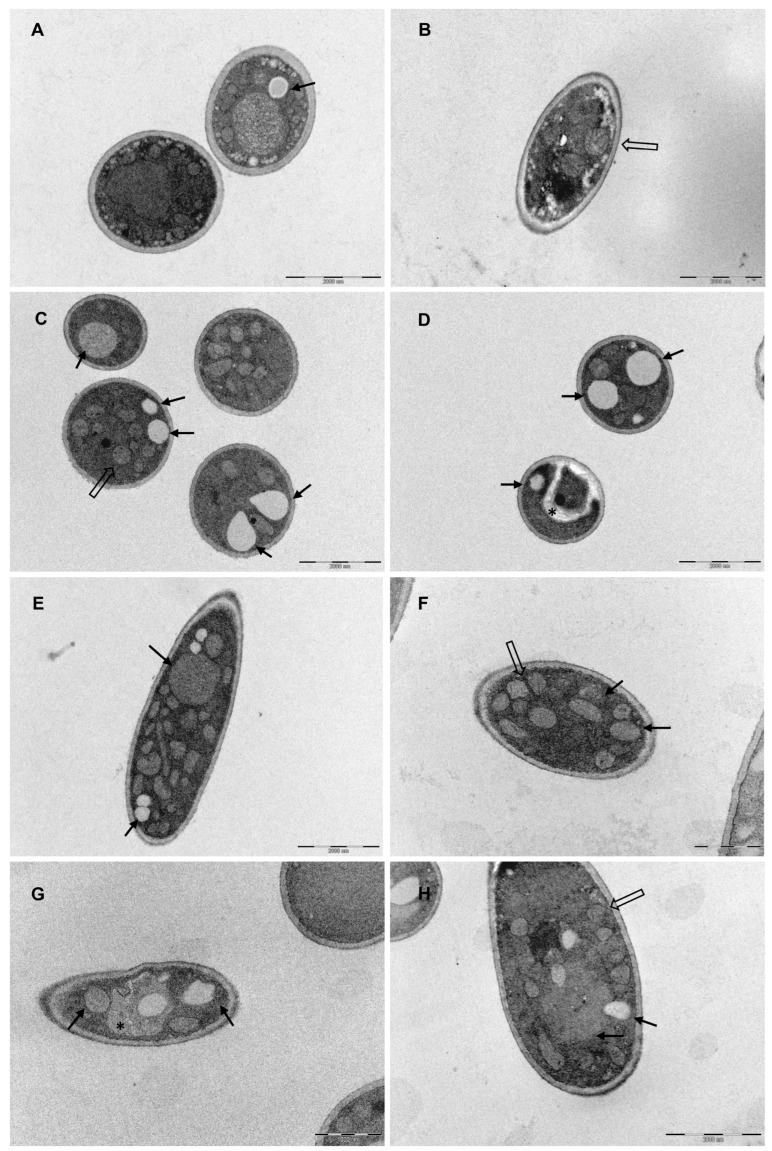
Transmission electron microscopic observations of *T. rubrum*. (**A**,**B**) Non-treated fungal cells, scale bar: 2000 nm; (**C**–**E**) treated with caffeinated SCG extract (137.5 μg/mL), scale bar: 2000 nm; (**F**–**H**) treated with decaffeinated SCG extract (150 μg/mL), (**F**,**E**) scale bar: 1000 nm, (**H**) scale bar 2000 nm*—autophagic-like structures; black arrow—vacuoles; open arrows—mitochondria.

**Table 1 microorganisms-11-00242-t001:** Phenolic profile of caffeinated and decaffeinated SCG ethanol extracts.

Peak	Rt (min)	λmax (nm)	[M-H]^−^ (*m/z*)	MS^2^ (*m/z*)	Tentative Identification	Quantification (mg/g Extract)		*t*-Students Test *p*-Value
						SCG Caffeinated	SCG Decaffeinated	
1	4.64	316	353	191(100), 179(78), 173(11), 135(14).	3-*O*-Caffeoylquinic acid	12.1 ± 0.1	8 ± 1	<0.001
2	5.23	319	353	191(49), 179(72), 173(100), 135(11).	4-*O*-Caffeoylquinic acid	5.26 ± 0.03	12.1 ± 0.3	<0.001
3	5.77	323	353	191(100), 179(49), 173(52), 135(12).	*cis* 5-*O*-Caffeoylquinic acid	16.6 ± 0.5	24 ± 1	<0.001
4	6.44	323	353	191(100), 179(28), 173(35), 135(5).	trans 5-*O*-Caffeoylquinic acid	24 ± 1	14 ± 1	<0.001
5	9.66	325	367	193(23), 191(5), 173(100), 161(10), 149(4).	4-*O*-Feruloylquinic acid	21.17 ± 0.02	2.49 ± 0.04	<0.001
6	12.01	326	335	179/13), 161(100), 134(41).	Caffeoylshikimic acid	17.2 ± 0.1	20.3 ± 0.5	<0.001
7	17.87	324	515	MS^2^: 353(100), 335(12), 317(5), 299(12), 203(10), 179(14), 173(13). MS^3^:191(52), 179(70), 173(100).	1,4-*O*-diCaffeoylquinic acid	11.3 ± 0.5	10.2 ± 0.3	<0.001
8	19.1	324	515	MS^2^: 353(100). MS^3^: 335(12), 191(100), 179(81), 173(142).	1,3-*O*-diCaffeoylquinic acid	7.02 ± 0.11	4.7 ± 0.2	<0.001
9	21.64	324	515	MS^2^: 353(100), 335(9), 317(5), 299(21), 203(9), 173(5). MS^3^:191(32), 179(51), 173(100).	3,4-*O*-diCaffeoylquinic acid	8.3 ± 0.2	6.8 ± 0.5	<0.001
10	23.27	325	529	MS^2^: 367(100), 335(20), 193(14). MS^3^: 173(100).	4-Feruloyl-5-caffeoylquinic acid	1.87 ± 0.03	1.5 ± 0.1	<0.001
11	27.29	325	529	MS^2^: 367(100), 193(14). MS^3^: 191(100).	3-Feruloyl-5-caffeoylquinic acid	2.1 ± 0.2	2.7 ± 0.1	<0.001
					Total Phenolic Compounds	127.39 ± 1.03	107 ± 1	<0.001

Standard calibration curves: chlorogenic acid (y = 168823x − 161172, *R*^2^ = 0.9999, LOD = 0.20 µg/mL; LOQ = 0.68 µg/mL, peaks 1, 2, 3, 4, 5, 6, 7, 8, and 9); ferulic acid (y = 633126x − 185462, *R*^2^ = 0.999; LOD = 0.20 μg/mL; LOQ = 1.01 μg/mL, peaks 5, 10, and 11).

**Table 2 microorganisms-11-00242-t002:** Minimum inhibitory concentration (MIC) of caffeinated and decaffeinated extracts and itraconazole for different species (ug/mL). Three independent experiments were performed.

Species	Caffeinated Coffee Extract		Decaffeinated Coffee Extract		Itraconazole
	GC	SCG	GC	SCG	
*A. alternata*	>1770.00	>1100.00	>1415.00	>2400.00	0.50
*A. fumigatus*	>1770.00	>1100.00	>1415.00	>2400.00	2.00
*A. infectoria*	>1770.00	>1100.00	>1415.00	>2400.00	0.50
*A. niger*	>1770.00	>1100.00	>1415.00	>2400.00	>16.00
*C. albicans*	>1770.00	>1100.00	>1415.00	>2400.00	0.06
*C. glabrata*	>1770.00	>1100.00	>1415.00	>2400.00	0.01
*C. krusei*	55.31	137.50	44.22	150.00	0.25
*C. parapsilosis*	110.63	275.00	44.22	150.00	0.03
*F. oxysporum*	>1770.00	>1100.00	>1415.00	>2400.00	>16.00
*T. mentagrophytes*	27.66	137.50	22.11	300.00	0.03
*T. rubrum*	27.66	137.50	22.11	150.00	0.03

**Table 3 microorganisms-11-00242-t003:** Minimum fungicidal concentration of caffeinated and decaffeinated SCG extracts against *T. mentagrophytes* and *T. rubrum*. Three independent experiments were performed.

Species	MFC (μg/mL)	
	Caffeinated SCG Extract	Decaffeinated SCG Extract
*T. mentagrophytes*	137.50	300
*T. rubrum*	137.50	150

**Table 4 microorganisms-11-00242-t004:** Cytotoxic and anti-inflammatory activity of the caffeinated and decaffeinated SCG extracts.

	Caffeinated SCG Extract	Decaffeinated SCG Extract	Positive Control
Cytotoxic activity (GI_50_ and µg/mL)			Ellipticine
AGS	55 ± 4	52 ± 2	1.20 ± 0.03
CaCo2	214 ± 9	179 ± 10	1.20 ± 0.02
MCF-7	207 ± 16	75.7 ± 0.3	1.00 ± 0.02
NCI-H460	217 ± 18	228 ± 10	1.21 ± 0.02
Cytotoxic activity (GI_50_ and µg/mL)			
PLP2	>400	240 ± 25	1.4 ± 0.1
Anti-inflammatory activity (IC_50_ and µg/mL)			Dexametasone
RAW 264.7	>400	>400	6.3 ± 0.4

## Data Availability

Not applicable.

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
