# Peer review of "Antifungal Activity of Spent Coffee Ground Extracts"

_microorganisms, 2023, doi:10.3390/microorganisms11020242_

Round 1
Reviewer 1 Report
The paper by Calheiros et al describes an interesting use for spent coffee ground extracts, namely antifungal activity, that could be employed in sustainable products such as cosmetics for instance. The study is well-conducted but there are details that need to be better explained. The use CG extracts as control is not clear, since it was only used in an experiment (antimicrobial susceptibility testing). The determination of citotoxicity and anti-inflammatory activity enrich the paper, but the results are not discussed and integrated with the results of antifungal activity. It would also enrich the paper to show results regarding the antioxidant activity of SGC extracts, since new ingredients to be used in cosmetics, determination of antifungal and antioxidant activity are key. Other minor comments are showed below.
line 61: meaning not clear- "Coffee is one most traded product"
line 119: "dried SCG or GC" - what "GC" stands for? Also appears as "CG" - I assume it was the control, from unused capsules but it is not clear on the text
line 231: please correct "at Analysis of phenolic compounds"
line 332: The MIC determined for all species are within de range of MIC breakpoints defined to be susceptible species according to CLSI protocols, except for Aspergillus spp. and F. oxysporum. However, probably due to strain- specific factors the concentration tested wasn’t high enough to allow MIC determination - please clarify this idea. Are the breakpoints defined for Aspergillus spp. and F. oxysporum? Therefore, an MIC > 1 ug/mL of itraconazole for these species is classified as resistant? If so,this reviewer agrees that there is no need to test a higher concentration. If not, then the MIC should be determined, as recommended by the CLSI guideline.
line 337: since the CG extract was more efficient, it is not clear why the MFC was determined only for SCG extracts.
line 458: What do the authors mean with "antioxidant activity" on Table 4?
line 477: microdilution broth is a technique and not a method. I suggest correcting to "antimicrobial susceptibility testing using the microdilution broth technique"
line 508: "pointing that despite less concentrated the bioactive compounds with antifungal activity remain present in SCG" - this information is not detailed in the manuscript. It is not clear to this reviewer why the extraction from the unused coffee capsules was not treated in the same way as the SCG extracts. It was only used as a control for the MIC determinations. The remaning assays did not include these extracts - not even the determination of the phenolic profile, which could add some valuable information, leading to the conclusion . I suggest removing these results since they do not add any substance to the manuscript.
line 516: Most antifungal therapy available targets these components, however, conventional antifungals are associated with many side effects and drug to drug interactions, being hepatotoxicity, nephrotoxicity, and myelotoxicity the most severe side effects - please discuss the citotoxicity and anti-inflammatory results and integrate them with the antifungal activity findings.
Author Response
Response to reviewers
The authors acknowledge the comments and questions of the reviewers that give us the opportunity to make modifications in the manuscript that increased its quality and clarity. A response point by point is given below and whenever needed it is mentioned the Line number of the Revised Version (Marked version):
Reviewer #1
Comments and Suggestions for Authors
The paper by Calheiros et al describes an interesting use for spent coffee ground extracts, namely antifungal activity, that could be employed in sustainable products such as cosmetics for instance. The study is well-conducted but there are details that need to be better explained. The use CG extracts as control is not clear, since it was only used in an experiment (antimicrobial susceptibility testing). The determination of citotoxicity and anti-inflammatory activity enrich the paper, but the results are not discussed and integrated with the results of antifungal activity. It would also enrich the paper to show results regarding the antioxidant activity of SGC extracts, since new ingredients to be used in cosmetics, determination of antifungal and antioxidant activity are key. Other minor comments are showed below.
line 61: meaning not clear- "Coffee is one most traded product"
The authors agree and changed the sentence to " Coffee is one most consumed beverage and one of the most traded commodities in the world".
line 119: "dried SCG or GC" - what "GC" stands for? Also appears as "CG" - I assume it was the control, from unused capsules but it is not clear on the text
SCG stands for Spent Coffee Ground (Line 14, 77) and GC for Ground Coffee (Line 122), and now, added in the line 121. We agree that CG was not correct and was changed to GC throughout the manuscript.
line 231: please correct "at Analysis of phenolic compounds"
Removed from line 237.
line 332: The MIC determined for all species are within de range of MIC breakpoints defined to be susceptible species according to CLSI protocols, except for Aspergillus spp. and F. oxysporum. However, probably due to strain- specific factors the concentration tested wasn’t high enough to allow MIC determination - please clarify this idea. Are the breakpoints defined for Aspergillus spp. and F. oxysporum? Therefore, an MIC > 1 ug/mL of itraconazole for these species is classified as resistant? If so,this reviewer agrees that there is no need to test a higher concentration. If not, then the MIC should be determined, as recommended by the CLSI guideline.
We thank the reviewer for this question that give us the opportunity to make additional antimicrobial tests with itraconazole for Aspergillus and Fusarium. We were able to correct the MIC values and to classify the fungal strains in terms of susceptibility to itraconazole.
Although the CLSI M38-A2 guidelines do not define breakpoints for mould testing, working breakpoints were assigned for analytical purposes. So, we determined the MIC of itraconazole at higher concentrations range for A. fumigatus and A. niger, F. oxysporum. We obtained that the A. fumigatus strain used is intermediate sensibility to itraconazole (MIC = 2 µg/mL) and A. niger and F. oxysporum are resistant (MIC > 16 µg/mL). We introduced these data in the Table 2.
The CLSI working breakpoints for analytical purposes, isolates were grouped as susceptible (MIC or MEC <1 μg/mL), intermediate (MIC or MEC 2 μg/mL), and resistant (MIC or MEC> 4 μg/mL).
Accordingly, the sentence beginning in Line 349 (line numbering of the revised version):
“A control was performed with itraconazole. The MIC determined for all species are within de range of MIC breakpoints defined to be susceptible species according to CLSI protocols, except for Aspergillus spp. and F. oxysporum. However, probably due to strain-specific factors the concentration tested wasn’t high enough to allow MIC determination.”
Was changed to:
“Although the CLSI M38-A2 guidelines do not define breakpoints for mould testing, working breakpoints were assigned for analytical purposes. Accordingly, the MIC determined for all the species tested are within the range of MIC breakpoints defined to be susceptible species, except for A. fumigatus that is considered intermediate and A. niger and F. oxysporum that are considered resistant.”
line 337: since the CG extract was more efficient, it is not clear why the MFC was determined only for SCG extracts.
The objective of this work was to study the antifungal activity of SCG, spent coffee grounds, in order to identify a way to recycle coffee waste. During the design of the work we felt that, as a control, it was important to study the antifungal effect of the same coffee grounds, (the same origin, the same coffee capsules) but unused (not brewed). This proved that the commercial coffee grounds inside the capsules had antifungal effect and that this effect, although lower, was kept after preparation of the coffee, excluding that the preparation of the beverage with the recommended coffee machine did not eliminated the bioactive compounds.
In order to clarify this aspect the sentence beginning in line 326 was changed to:
“Although the aim of this work was to study the antifungal activity of SCG, we used as a control, the same coffee grounds, same origin, same coffee capsules, but not previously brewed. The objective of using GC was to verify if the commercial coffee grounds inside the capsules had antifungal effect, and if the brewing, with the recommended coffee machine, did not eliminated the bioactivity. The results showed that GC had a stronger fungal growth inhibition than SCG, with lower MIC values when compared with SCG extracts (Table 2).”
line 458: What do the authors mean with "antioxidant activity" on Table 4?
We thank the reviewer for this comment. “Antioxidant activity” was removed from Table 2.
line 477: microdilution broth is a technique and not a method. I suggest correcting to "antimicrobial susceptibility testing using the microdilution broth technique"
The designation of the methodology was changed as follows:
” Broth microdilution was performed to determine the Minimum Inhibitory Concentration (MIC) of the extracts, following the standard protocol M27-A2 for yeasts and the M38-A2 protocol for filamentous fungi (CLSI).”
Was changed to:
“antimicrobial susceptibility testing using the microdilution broth technique was performed to determine the Minimum Inhibitory Concentration (MIC) of the extracts, following the standard protocol M27-A2 for yeasts and the M38-A2 protocol for filamentous fungi (CLSI).”.
We also changed in line 151:
“2.4.1. Microdilution broth”,
to:
“2.4.1. Antimicrobial susceptibility testing”
and in line 322:
“Microdilution broth assay”
to
“Antimicrobial susceptibility testing”
line 508: "pointing that despite less concentrated the bioactive compounds with antifungal activity remain present in SCG" - this information is not detailed in the manuscript. It is not clear to this reviewer why the extraction from the unused coffee capsules was not treated in the same way as the SCG extracts. It was only used as a control for the MIC determinations. The remaning assays did not include these extracts - not even the determination of the phenolic profile, which could add some valuable information, leading to the conclusion . I suggest removing these results since they do not add any substance to the manuscript.
We thank the reviewer for raising this comment that is an opportunity to explain and to increase the clarity of the work. As already mentioned in this response to reviewer (Response to line 337) the sentence beginning in Line 326 (revised version) was changed to
“Although the aim of this work was to study the antifungal activity of SCG, we used as a control, the same coffee grounds, same origin, same coffee capsules, but not previously brewed. The objective of using GC was to verify if the commercial coffee grounds inside the capsules had antifungal effect, and if the brewing, with the recommended coffee machine, did not eliminated the bioactivity. The results showed that GC had a stronger fungal growth inhibition than SCG, with lower MIC values when compared with SCG extracts (Table 2).”
Line 527 (Discussion) was also changed from:
“Control using ground coffee (GC) extract were also tested in the same species to compare the antifungal potential between GC and SCG. As expected, GC have stronger antifungal activity, exhibiting lower MIC values. The brewing process of GC removes water- soluble compounds making SCG less concentrated, nonetheless SCG extracts have antifungal activity against the same species that GC extracts, pointing that despite less concentrated the bioactive compounds with antifungal activity remain present in SCG.”
To
“Control using ground coffee (GC) extract revealed that unused coffee grounds had antifungal activity, with lower MIC values. The brewing process of GC removes water- soluble compounds making SCG less concentrated, nonetheless SCG extracts have antifungal activity against the same species that GC extracts, pointing that despite less concentrated the bioactive compounds with antifungal activity remain present in SCG.”
However, if the reviewer thinks that this is negligible, we will remove it from the results.
line 516: Most antifungal therapy available targets these components, however, conventional antifungals are associated with many side effects and drug to drug interactions, being hepatotoxicity, nephrotoxicity, and myelotoxicity the most severe side effects - please discuss the citotoxicity and anti-inflammatory results and integrate them with the antifungal activity findings.
The authors thank the reviewer for this recommendation. The following text was introduced (Line 592):
“In this study, we also reported that the SCG caffeinated and decaffeinated coffee extracts are cytotoxic for different cancer cell lines, without causing cytoxicity on the non-tumoral cell line used. This is in accordance with previous works that reports a moderate cytotoxicity of SCG extracts on liver cancer cells (Hep-G2) also lower than the positive control used in the assay [36]. However, our results did not show anti-inflammatory activity of SCG caffeinated and decaffeinated coffee extracts on RAW 264.7 cell line. A recent publication reported that the SCG water extracts have more effective than extracts obtained with solvents [49]. Nevertheless, when considering the antifungal activity of the extracts obtained in this study in fungi agents of human skin infection together with the absence of cytotoxicity on a non-tumoral cell line, it can be anticipated a possible utilization in the therapy of dermatomycosis..”
List of references
- Badr AN, El-Attar MM, Ali HS, Elkhadragy MF, Yehia HM, Farouk A. Spent Coffee Grounds Valorization as Bioactive Phenolic Source Acquired Antifungal, Anti-Mycotoxigenic, and Anti-Cytotoxic Activities. Toxins. 2022; 14(2):109. https://doi.org/10.3390/toxins14020109.
- Angeloni, S., Freschi, M., Marrazzo, P., Hrelia, S., Beghelli, D., Juan-García, A., Juan, C., Caprioli, G., Sagratini, G., & Angeloni, C. (2021). Antioxidant and Anti-Inflammatory Profiles of Spent Coffee Ground Extracts for the Treatment of Neurodegener-ation. Oxidative medicine and cellular longevity, 2021, 6620913.
Reviewer 2 Report
In this manuscript, the antifungal activity of ethanolic SCG extracts from caffeinated and decaffeinated coffee capsules was evaluated against yeasts and filamentous fungi,
so as to turn waste into wealth. It is an interesting work, and the experiment is well design. However, there are still several critical points needing to be clarified.
1. L104, L108: ……were inoculated and incubated at 30℃ for how long time?
2. L313-: There seems no the description about the difference between the antifungal activity of caffeinated and decaffeinated extracts?
3. L410-412: There appear to be some structures that cross the cytoplasmic membrane and the cell wall to the outside, which was observed in C. parapsilosis untreated cells. Maybe, the treatment method of cell sample for TEM may need to be considered.
4. In the discussion, there is no need to repeat the results in detail.
5. Compared with other reports, how is antifungal effect of coffee extract in this study. Some comparative analysis should be added in the discussion.
6. The discussion on cytotoxicity activity and anti-inflammatory activity of coffee extract in this study is lacking.
7. In this study, some functions of ethanolic caffeinated and decaffeinated SCG extracts was evaluated, but the Conclusions need to be specific and clear.
Author Response
Response to reviewers
The authors acknowledge the comments and questions of the reviewers that give us the opportunity to make modifications in the manuscript that increased its quality and clarity. A response point by point is given below and whenever needed it is mentioned the line number of the revised version (Marked version):
Reviewer #2
Comments and Suggestions for Authors
In this manuscript, the antifungal activity of ethanolic SCG extracts from caffeinated and decaffeinated coffee capsules was evaluated against yeasts and filamentous fungi,
so as to turn waste into wealth. It is an interesting work, and the experiment is well design. However, there are still several critical points needing to be clarified.
- L104, L108: ……were inoculated and incubated at 30℃ for how long time?
For the yeast, we added “, for 24h”, Line 108
For the Filamentous fungi, we added “, for a week”, Line 111 and 112.
- L313-: There seems no the description about the difference between the antifungal activity of caffeinated and decaffeinated extracts?
We acknowledge the reviewers comment. In fact there was no description of the differences between caffeinated and decaffeinated extracts. To correct this, the first paragraph of the subsection 3.2 Antifungal activity (Line 321), was changed to:
“Antimicrobial susceptibility testing was performed to determine MIC of the SCG and GC extracts, caffeinated and decaffeinated, following standardized CLSI protocol M27-A2 for yeasts and M38-A2 for filamentous fungi. Aqueous extracts of SCG did not show growth inhibition in all the species tested, so there were no further used during the present work. Although the aim of this work was to study the antifungal activity of SCG, we used as a control, the same coffee grounds, same origin, same coffee capsules, but unused. The objective of using GC was to verify if the commercial coffee grounds inside the capsules had antifungal effect, and if the brewing, with the recommended coffee machine did not eliminated the bioactivity. The results showed that GC had a stronger fungal growth inhibition than SCG, with lower MIC values when compared with SCG extracts (Table 2). The species C. krusei, C. parapsilosis, T. mentagrophytes and T. rubrum are the most susceptible to the tested extracts. The SCG did not show antifungal activity in the following species: A. alternata, A. infectoria, A. fumigatus, A. niger, C. albicans, C. glabrata and F. oxysporum, so the following studies only proceeded on the most susceptible species.The most susceptible species to decaffeinated extract were also C. krusei, C. parapsilosis, T. mentagrophytes and T. rubrum. While the decaffeinated SCG MIC values here higher than the MIC values for caffeinated SCG, the opposite was obtained for GC extracts (Table 2). “
- L410-412: There appear to be some structures that cross the cytoplasmic membrane and the cell wall to the outside, which was observed inC. parapsilosis untreated cells. Maybe, the treatment method of cell sample for TEM may need to be considered.
In fact, we are not sure about this structure. But it strike our attention because, regardless of using exactly the same methodology to prepare TEM samples, these structures were only observed for C. parapsilosis - a species specific structure/modification of the surface decoration/architecture? We described this in more detail because we believe that this might be involved in the response to these extracts since there is an increase in the number of structures when C. parapsilosis is in the presence of the extract; also, this might be an important information for other colleagues studying C. parapsilosis. We did not add any other comment in the manuscript because this is an observation that deserves further insights.
- In the discussion, there is no need to repeat the results in detail.
The paragraph beginning (Line 558) with “Quantification of ergosterol, β-(1,3)-glucan and chitin after exposure to the MIC of SCG from caffeinated and decaffeinated coffee extracts was performed. SCG caffeinated extract induced significant impairment of almost 50% on ergosterol, 14.83% on β-(1,3)-glucan and 21.66% on chitin of C. parapsilosis, while SCG from decaffeinated, caused a non-significant reduction in ergosterol and β-(1,3)-glucan levels but a significant decrease in chitin content.”
Was changed to (Line 558): “Quantification of ergosterol, β-(1,3)-glucan and chitin after exposure to the MIC of SCG from caffeinated and decaffeinated coffee extracts was performed. SCG caffeinated extract induced significant impairment on ergosterol, β-(1,3)-glucan and chitin contents of C. parapsilosis, while SCG from decaffeinated, caused a non-significant reduction in ergosterol and β-(1,3)-glucan levels but a significant decrease in chitin content.”
Also, in Line 579, it was removed the “25.69%”.
- Compared with other reports, how is antifungal effect of coffee extract in this study. Some comparative analysis should be added in the discussion.
We thank the reviewer for raising the lack of discussion about other studies showing the antifungal effect of SCG. We added this to the discussion and several references. There aren’t many studies showing the antifungal effect of spent coffee. Most of the works have a focus in bacteria, especially related to food infection.
In Line 538 we added the following:
“Previous studies have highlighted the antimicrobial activity of SCG, mostly with a focus in bacteria [33-37]. Studies describing the antifungal effect of SCG are scarce. Several yeast species (C. krusei, C. parapsilosis and C. albicans) have al-ready been reported to be susceptible to SCG extracts [12, 34]. The antifungal activity against filamentous fungi was also recently reported for a SCG extract obtained using isopropanol [36]. However it is difficult to compare between studies due to different methodologies (type of coffee, extraction method, antifungal susceptibility tests, fungal strains). Regardless, this is the first study describing the antifungal effect of SCG against fungal agents of dermatophytosis, T. mentagrophytes and T. rubrum. ”
Reference list:
- Sousa, C.; Gabriel, C.; Cerqueira, F.; Manso, M. C.; Vinha, A. F. Coffee industrial waste as a natural source of bioactive com-pounds with antibacterial and antifungal activities. In The Battle Against Microbial Pathogens: Basic Science, Technological Advances and Educational Programs, Méndez-Vilas, A., Ed., 2016, 131–136.
- Almeida, A. A. P., Farah, A., Silva, D. A., Nunan, E. A., & Glória, M. B. A..Antibacterial activity of coffee extracts and selected coffee chemical compounds against enterobacteria. Journal of agricultural and food chemistry, 2006, 54(23), 8738-8743.
- Monente, C., Bravo, J., Vitas, A. I., Arbillaga, L., De Peña, M. P., & Cid, C.. Coffee and spent coffee extracts protect against cell mutagens and inhibit growth of food-borne pathogen microorganisms. Journal of Functional Foods, 2015, 12, 365-374.
- Díaz-Hernández, G.C.; Alvarez-Fitz, P.; Maldonado-Astudillo, Y.I.; Jiménez-Hernández, J.; Parra-Rojas, I.; Flores-Alfaro, E.; Salazar, R.; Ramírez, M. Antibacterial, Antiradical and Antiproliferative Potential of Green, Roasted, and Spent Coffee Ex-tracts. Appl. Sci. 2022, 12, 1938. https://doi.org/10.3390/app12041938.
- Badr AN, El-Attar MM, Ali HS, Elkhadragy MF, Yehia HM, Farouk A. Spent Coffee Grounds Valorization as Bioactive Phenolic Source Acquired Antifungal, Anti-Mycotoxigenic, and Anti-Cytotoxic Activities. Toxins. 2022; 14(2):109. https://doi.org/10.3390/toxins14020109.
- Rawangkan, A., Siriphap, A., Yosboonruang, A., Kiddee, A., Pook-In, G., Saokaew, S., Sutheinkul, O., & Duangjai, A. Potential Antimicrobial Properties of Coffee Beans and Coffee By-Products Against Drug-Resistant Vibrio cholerae. Frontiers in nutri-tion, ,2022, 9, 865684. https://doi.org/10.3389/fnut.2022.865684
- The discussion on cytotoxicity activity and anti-inflammatory activity of coffee extract in this study is lacking.
The authors totally agree with the reviewer comment, also raised by other reviewer, and, in order to discuss this important aspect of our results, added the following (Line 587):
“In this study, we also reported that the SCG caffeinated and decaffeinated coffee extracts are cytotoxic for different cancer cell lines, without causing cytoxicity on the non-tumoral cell line used. This is in accordance with previous works that reports a moderate cytotoxicity of SCG extracts on liver cancer cells (Hep-G2) also lower than the positive control used in the assay [36]. However, our results did not show anti-inflammatory activity of SCG caffeinated and decaffeinated coffee extracts on RAW 264.7 cell line. A recent publication reported that the SCG water extracts have more effective than extracts obtained with solvents [49].”
- Badr AN, El-Attar MM, Ali HS, Elkhadragy MF, Yehia HM, Farouk A. Spent Coffee Grounds Valorization as Bioactive Phenolic Source Acquired Antifungal, Anti-Mycotoxigenic, and Anti-Cytotoxic Activities. Toxins. 2022; 14(2):109. https://doi.org/10.3390/toxins14020109
- Angeloni S, Freschi M, Marrazzo P, Hrelia S, Beghelli D, Juan-García A, Juan C, Caprioli G, Sagratini G, Angeloni C. Antioxidant and Anti-Inflammatory Profiles of Spent Coffee Ground Extracts for the Treatment of Neurodegeneration. Oxid Med Cell Longev. 2021 May 19;2021:6620913. doi: 10.1155/2021/6620913.
- In this study, some functions of ethanolic caffeinated and decaffeinated SCG extracts was evaluated, but the Conclusions need to be specific and clear.
The conclusions were changed to:
“Coffee consumption generates tons of solid waste SCG, containing several bioactive compounds with potential uses in pharmaceutical, food and cosmetic industries [16]. In this study we show that the ethanolic SCG extracts of both caffeinated and decaffeinated coffee are cytotoxic for different cancer cell lines, indicating antiproliferative bioactivity, with no cytoxicity on a non-tumoral cell line . Our results also clearly demonstrate that these extracts have a strong anti-fungal bioactivity, especially against fungi involved in skin infections. This indicates that a safe formulation of a SCG based phytotherapeutic for topic administration could be an economic and eco-friendly alternative to treat skin fungal infections, promoting a coffee by-product and minimizing waste.”
Reviewer 3 Report
The current work focuses on the antifungal activity of spent coffee grounds extracts. The experimental work appears to have been carried out well. However, a few points deserve attention for further publication. I suggest that it is accepted for publication after the following revisions:
1) Once you cite the name of the species for the first time, you need to show it in full length. In all other instances, you need to make use of the contracted genus name. Please check this in the manuscript.
2) Page 2, line 66: The previous studies on the antifungal properties of coffee should be discussed in the manuscript.
3) Page 3, line 112; Page 6, line 294; Page 8, line 314; Table 2; Page 9, line 346; Page 10, line 395: There are some confusing abbreviations in these places, GC or CG, SGC or SCG. GC or CG should also be explained in the introduction or methods.
4) Page 3, line 150: Add the full name of CLSI.
5) Page 5, line 231: Delete: at Analysis of phenolic compounds.
6) Please check the statements in the manuscript carefully. Page 3, line 122 and line 124; Page 6, line 297 and line 299: extraction with water and ethanol/water 70:30?
7) Table 2: Change "Caffeinated SCG extract" to " Caffeinated coffee extract", change "Decaffeinated SCG extract" to " Decaffeinated coffee extract".
8) Page 8, line 313: It would be more interesting if more in-depth discussion on the comparative inhibition of caffeinated and decaffeinated SCG extracts.
9) Page 10, line 384: The manuscript will be more readable if the authors add a brief summary of this experiment, as well as the ideas and purposes of the next experiment.
Author Response
Response to reviewers
The authors acknowledge the comments and questions of the reviewers that give us the opportunity to make modifications in the manuscript that increased its quality and clarity. A response point by point is given below and whenever needed it is mentioned the line number of the revised version (Marked version):
Reviewer #3
Comments and Suggestions for Authors
The current work focuses on the antifungal activity of spent coffee grounds extracts. The experimental work appears to have been carried out well. However, a few points deserve attention for further publication. I suggest that it is accepted for publication after the following revisions:
1) Once you cite the name of the species for the first time, you need to show it in full length. In all other instances, you need to make use of the contracted genus name. Please check this in the manuscript.
The entire manuscript was revised and changes introduced whenever needed to comply with this rule.
2) Page 2, line 66: The previous studies on the antifungal properties of coffee should be discussed in the manuscript.
This same question was raised by reviewer #2. We acknowledge that the reviewers made this comment since it give us the opportunity to increase the quality of the discussion. In Line 538 (revised version) we added the following:
“Previous studies have highlighted the antimicrobial activity of SCG, mostly with a focus in bacteria [33-37]. Studies describing the antifungal effect of SCG are scarce. Several yeast species (C. krusei, C. parapsilosis and C. albicans) have al-ready been reported to be susceptible to SCG extracts [12, 34]. The antifungal activity against filamentous fungi was also recently reported for a SCG extract obtained using isopropanol [36]. However it is difficult to compare between studies due to different methodologies (type of coffee, extraction method, antifungal susceptibility tests, fungal strains). Regardless, this is the first study describing the antifungal effect of SCG against fungal agents of dermatophytosis, T. mentagrophytes and T. rubrum. ”
Reference list:
- Sousa, C.; Gabriel, C.; Cerqueira, F.; Manso, M. C.; Vinha, A. F. Coffee industrial waste as a natural source of bioactive com-pounds with antibacterial and antifungal activities. In The Battle Against Microbial Pathogens: Basic Science, Technological Advances and Educational Programs, Méndez-Vilas, A., Ed., 2016, 131–136.
- Almeida, A. A. P., Farah, A., Silva, D. A., Nunan, E. A., & Glória, M. B. A..Antibacterial activity of coffee extracts and selected coffee chemical compounds against enterobacteria. Journal of agricultural and food chemistry, 2006, 54(23), 8738-8743.
- Monente, C., Bravo, J., Vitas, A. I., Arbillaga, L., De Peña, M. P., & Cid, C.. Coffee and spent coffee extracts protect against cell mutagens and inhibit growth of food-borne pathogen microorganisms. Journal of Functional Foods, 2015, 12, 365-374.
- Díaz-Hernández, G.C.; Alvarez-Fitz, P.; Maldonado-Astudillo, Y.I.; Jiménez-Hernández, J.; Parra-Rojas, I.; Flores-Alfaro, E.; Salazar, R.; Ramírez, M. Antibacterial, Antiradical and Antiproliferative Potential of Green, Roasted, and Spent Coffee Ex-tracts. Appl. Sci. 2022, 12, 1938. https://doi.org/10.3390/app12041938.
- Badr AN, El-Attar MM, Ali HS, Elkhadragy MF, Yehia HM, Farouk A. Spent Coffee Grounds Valorization as Bioactive Phenolic Source Acquired Antifungal, Anti-Mycotoxigenic, and Anti-Cytotoxic Activities. Toxins. 2022; 14(2):109. https://doi.org/10.3390/toxins14020109.
- Rawangkan, A., Siriphap, A., Yosboonruang, A., Kiddee, A., Pook-In, G., Saokaew, S., Sutheinkul, O., & Duangjai, A. Potential Antimicrobial Properties of Coffee Beans and Coffee By-Products Against Drug-Resistant Vibrio cholerae. Frontiers in nutri-tion, ,2022, 9, 865684. https://doi.org/10.3389/fnut.2022.865684
3) Page 3, line 112; Page 6, line 294; Page 8, line 314; Table 2; Page 9, line 346; Page 10, line 395: There are some confusing abbreviations in these places, GC or CG, SGC or SCG. GC or CG should also be explained in the introduction or methods.
Throughout the entire manuscript we corrected the designations SCG and CG.
4) Page 3, line 150: Add the full name of CLSI.
Changed accordingly, Clinical and Laboratory Standards Institute was added to Line 155.
5) Page 5, line 231: Delete: at Analysis of phenolic compounds.
Deleted accordingly.
6) Please check the statements in the manuscript carefully. Page 3, line 122 and line 124; Page 6, line 297 and line 299: extraction with water and ethanol/water 70:30?
These lines were carefully revised and modifications were introduced:
Line 125: we removed “water and”.
Line 127: we removed “with water”
Line 305 and 307: it was removed “and water”.
7) Table 2: Change "Caffeinated SCG extract" to " Caffeinated coffee extract", change "Decaffeinated SCG extract" to " Decaffeinated coffee extract".
Changed accordingly
8) Page 8, line 313: It would be more interesting if more in-depth discussion on the comparative inhibition of caffeinated and decaffeinated SCG extracts.
To correct this, the first paragraph of the subsection 3.2 Antifungal activity, was changed to:
“Antimicrobial susceptibility testing was performed to determine MIC of the SCG and GC extracts, caffeinated and decaffeinated, following standardized CLSI protocol M27-A2 for yeasts and M38-A2 for filamentous fungi. Aqueous extracts of SCG did not show growth inhibition in all the species tested, so there were no further used during the present work. Although the aim of this work was to study the antifungal activity of SCG, we used as a control, the same coffee grounds, same origin, same coffee capsules, but unused. The objective of using GC was to verify if the commercial coffee grounds inside the capsules had antifungal effect, and if the brewing, with the recommended coffee machine did not eliminated the bioactivity. The results showed that GC had a stronger fungal growth inhibition than SCG, with lower MIC values when compared with SCG extracts (Table 2). The species C. krusei, C. parapsilosis, T. mentagrophytes and T. rubrum are the most susceptible to the tested extracts. The SCG did not show antifungal activity in the following species: A. alternata, A. infectoria, A. fumigatus, A. niger, C. albicans, C. glabrata and F. oxysporum, so the following studies only proceeded on the most susceptible species.The most susceptible species to decaffeinated extract were also C. krusei, C. parapsilosis, T. mentagrophytes and T. rubrum. While the decaffeinated SCG MIC values here higher than the MIC values for caffeinated SCG, the opposite was obtained for GC extracts (Table 2). “
9) Page 10, line 384: The manuscript will be more readable if the authors add a brief summary of this experiment, as well as the ideas and purposes of the next experiment.
Since there are different line numberings in the pdf and the Word document withdrawn from we were not sure about the exact experiment the reviewer was referring to. Nevertheless, we interpreted this question regarding the description of the rationale beyond the experiments to introduce the results obtained (Page 10, Results section).
In fact, we initiate which paragraph with an explanation of the objective of the experiments, for example:
Line 358: “To understand whether the effect of the extracts is fungicidal or fungistatic against the most susceptible species, MFC was determined for SCG extracts”
Line 369: “To elucidate a possible mechanism of action underlying the antifungal effect, er-gosterol, β-(1,3)-glucan and chitin contents of C. krusei, C. parapsilosis, T. mentagrophytes and T. rubrum were determined after exposure to MIC of SCG extracts (caffeinated and decaffeinated) and expressed in percentage in comparison to non-treated fungal cells (control 100%). Ergosterol was also quantified after exposure to 0.016 μg/mL of itracon-azole as a positive control, a lower concentration than the MIC value was used to allow fungal growth and to have enough biomass to extract and quantify ergosterol.”
Nevertheless, we kindly ask the reviewer to elucidate if we are making a correct interpretation of this question.
In the pdf, the line 384 is empty and before the figure.
We were not able to see this aspect of the text
Round 2
Reviewer 3 Report
Although the authors have revised the manuscript followed the comments of reviewers, there is still a point which is not response by the authors, it need to be revised for further publication.
2) Page 2, line 66: The previous studies on the antifungal properties of coffee should be discussed in the manuscript.
Author Response
The focus of our study was the coffee ground spents, but we agree with the reviewer that introducing the overall potential of coffee and coffee fruit extracts as a source of antifungals can complement and enrich the manuscript. So, beginning in Line 72 we introduced the following text and 9 additional references:
“The coffee fruit has been indicated as having antimicrobial activities including against fungi. In fact, the fruit pulp (18,19,20), the skin (21) and the seed extracts (22,23), either roasted (24) and brewed (25,26) have been demonstrated to have bioactivity against fungi agents of animal/human infection and phytopathogens.”
- Martínez-Tomé, M., Jiménez-Monreal, A.M., García-Jiménez, Almela, L.L., García-Diz, L., Mariscal-Arcas, M. & Murcia, M.A. Assessment of antimicrobial activity of coffee brewed in three different ways from different origins. Eur Food Res Technol. 2011, 233, 497. https://doi.org/10.1007/s00217-011-1539-0
- Duangjai, A., Suphrom, N., Wungrath, J., Ontawong, A., Nuengchamnong, N. & Yosboonruang, A. Comparison of antioxidant, antimicrobial activities and chemical profiles of three coffee (Coffea arabica L.) pulp aqueous extracts. Integr Med Res. 2016 5(4):324-331. https://doi.org/10.1016/j.imr.2016.09.001.
- Alvarado-Ambriz, S., Lobato-Calleros, C., Hernández-Rodríguez, L., & Vernon-Carter, E.. Wet processing coffee waste as an alternative to produce extracts with antifungal activity: In vitro and in vivo valorization. Revista Mexicana De Ingeniería Química. 2020, 19(Sup. 1), 135-149. https://doi.org/10.24275/rmiq/Bio1612.
- Sangta, J., Wongkaew, M.,Tangpao, T.;Withee, P. Haituk, S. Arjin, C., Sringarm, K., Hongsibsong, S., Sutan, K., Pusadee, T., Sommano, S.R. & Cheewangkoon, R. Recovery of Polyphenolic Fraction from Arabica Coffee Pulp and Its Antifungal Applications. Plants. 2021, 10, 1422. https://doi.org/10.3390/plants10071422.
- Rodrigues F, Palmeira-de-Oliveira A, das Neves J, Sarmento B, Amaral MH, Oliveira MB. Coffee silverskin: a possible valuable cosmetic ingredient. Pharm Biol. 2015, 53(3), 386-94. https://doi.org/10.3109/13880209.2014.922589.
- Mathur, I., Shruthi, S., Gandrakota, K., & Nisha, K.K. Comparative Evaluation of Antifungal Activity of Green Coffee and Green Tea Extract against Candida albicans: An In Vitro Study. World J Dent 2021;12(4):265–270.
- Antoine, G., Vaissayre, V., Meile, J.-C., Payet, J., Conéjéro, G., Costet, L., Fock-Bastide, I., Joët, T. & Dussert, S. Diterpenes of Coffea seeds show antifungal and anti-insect activities and are transferred from the endosperm to the seedling after germination. Plant Physiology and Biochemistry. 2023, 194, 627-637.https://doi.org/10.1016/j.plaphy.2022.12.013.
- Mahajan, R. & Kapoor, N. Phytochemical analysis and antimicrobial activity of roasted beans of coffea robusta. International Journal of Pharmacy and Biological Sciences. 2018, 8, 89-95.
- Martínez-Tomé, M., Jiménez-Monreal, A.M., García-Jiménez, Almela, L.L., García-Diz, L., Mariscal-Arcas, M. & Murcia, M.A. Assessment of antimicrobial activity of coffee brewed in three different ways from different origins. Eur Food Res Technol. 2011, 233, 497. https://doi.org/10.1007/s00217-011-1539-0
- Mehta, V. Rajesh, G., Rao, A., Shenoy, R., Pai, M. Antimicrobial Efficacy of Punica granatum mesocarp, Nelumbo nucifera Leaf, Psidium guajava Leaf and Coffea Canephora Extract on Common Oral Pathogens: An In-vitro Study. Journal of Clinical and Diagnostic Research. 2014, 8, 65-68.